# Multi-Teacher Knowledge Distillation Augmented Group Relative Policy Optimization

## Abstract

Transfer learning, a key paradigm for leveraging pre-existing knowledge, can significantly enhance reinforcement learning agents, particularly when dealing with Large Language Models (LLMs) and Small Language Models (SLMs). Knowledge Distillation (KD) provides a potent mechanism for this transfer from expert LLM teacher models to SLM student models. Group Relative Policy Optimization (GRPO) is a robust critic-free reinforcement learning algorithm effective for policy optimization by estimating advantage via intra-group reward comparisons. Standard GRPO, however, does not inherently incorporate guidance from external expert policies and can exhibit training instability. This paper introduces a novel theoretical framework to integrate multi-teacher KD with GRPO. We propose a family of GRPO-KD objective functions; our primary formulation augments GRPO with an explicit, adaptively weighted multi-teacher distillation term to preserve stability for the SLM training. We further explore two advanced strategies: one modifying the Kullback-Leibler (KL) regularization of GRPO, and another introducing a Teacher Agreement Score to directly modulate the advantage calculation for deeper guidance from multiple LLM teachers. Experimental results on benchmark reasoning tasks demonstrate that the proposed framework not only stabilizes the training process but also significantly outperforms standard GRPO and other baseline approaches, validating the effectiveness of synergizing critic-free RL with multi-teacher guidance.

## 1 Introduction

The field of Natural Language Processing (NLP) has been fundamentally reshaped by the advent of Large Language Models (LLMs). These models, characterized by their extensive parameter counts and training on vast textual corpora, have demonstrated advanced capabilities in understanding, generating, and reasoning with human language (Raffel et al., 2020). Concurrently, Small Language Models (SLMs) have emerged as computationally efficient and often more specialized alternatives (Van Nguyen et al., 2024). SLMs are particularly valuable in resource-constrained environments or for tasks requiring rapid inference and customization (Sakib et al., 2025). They are frequently developed by distilling knowledge from larger teacher models or by fine-tuning pre-trained models of a more modest scale (Zhou et al., 2023; Ballout et al., 2024; Kang et al., 2023).

Reinforcement Learning (RL) has emerged as an important paradigm for fine-tuning language models to perform complex sequential decision-making tasks (Wang et al., 2024a). Proximal Policy Optimization (PPO) algorithms (Schulman et al., 2017), which directly search for an optimal policy, are central to many RL successes. However, challenges such as sample inefficiency (Havrilla et al., 2024; Zhao et al., 2023) and training instability persist (Zheng et al., 2023; Yuan et al., 2025), especially with high-dimensional policy spaces. Group Relative Policy Optimization (GRPO) (Shao et al., 2024) offers a critic-free approach that estimates advantages by comparing rewards of multiple candidate outputs, enhancing stability and efficiency.

Concurrently, Knowledge Distillation (KD) has been widely adopted for fine-tuning language models due to its stability, serving as a bridge to transfer knowledge, compress models, and improve performance in resource-constrained settings (Wen et al., 2025). This process typically involves training a student model to mimic the

soft probabilistic outputs or internal feature representations learned by a larger teacher model (Mansourian et al., 2025). However, limitations exist, namely transferring superficial patterns rather than deep reasoning capabilities (Wang et al., 2024b), the student model inheriting inaccuracies and hallucinations (Song et al., 2025), noisy student-generated outputs reducing generalization (Koo et al., 2025), homogenization (Lee et al., 2025), capacity gap (Zhang et al., 2024), and loss of specialized knowledge (Li et al., 2024b).

A deep, simultaneous blend of RL and TL offers a compelling path forward. Such synergy can leverage the strengths of both paradigms: transferred knowledge can guide RL exploration, making learning more sample-efficient and focused; it can inform reward shaping, providing denser or more meaningful signals; and RL can, in turn, dynamically select, modulate, or refine the transferred knowledge from KD, making it more relevant to the current task. Thus, this paper focuses on the theoretical integration of GRPO with guidance from multiple teacher policies leading to a family of KD-enhanced GRPO objective functions which we denote as GRPO-KD.

Ergo, the main contribution of this paper is the formalization of a theoretical basis extending GRPO with multi-teacher KD, where three objective functions are derived and evaluated on downstream reasoning tasks to fine-tune an SLM. We believe this is the first attempt at merging these two concepts together through the extension of the GRPO algorithm.

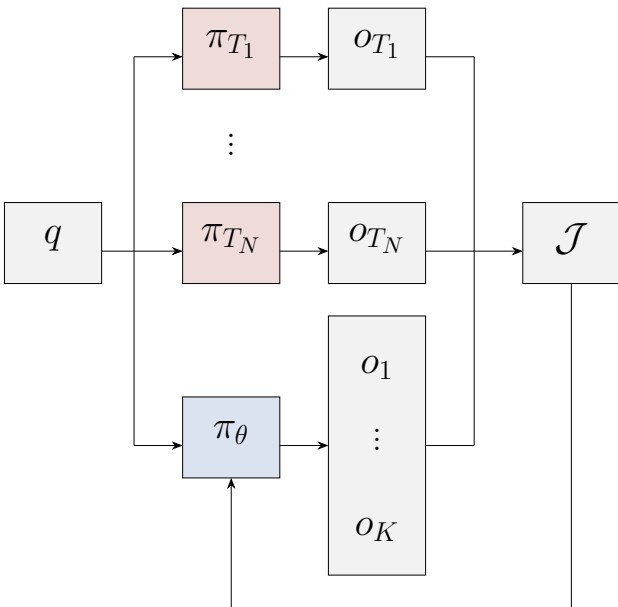

Figure 1: Overview of the proposed approach.

The general approach adopted within this paper is shown in Figure 1. This paper introduces a family of three novel objective functions, termed GRPO-KD, designed to integrate multi-teacher knowledge distillation (MTKD) with the GRPO framework. The first and primary objective, $\mathcal{J}'(\theta)$, augments the standard GRPO loss with a distinct, adaptively weighted KD term, preserving the modularity between relative reward optimization and teacher guidance. The second approach, $\mathcal{J}''(\theta)$, offers a tighter integration by replacing the standard GRPO Kullback-Leibler (KL) regularization with a Hinton KL regularization and the reference policy with a weighted composite of the teacher policies, thereby using the teachers to directly guide stabilization.

The third and final method, $\mathcal{J}'''(\theta)$, achieves the deepest integration by directly modifying the advantage calculation itself. It introduces a "Teacher Agreement Score" to augment the group-relative advantage, effectively embedding the consensus of the teacher ensemble into the core learning signal of the RL agent. Experimental results on benchmark reasoning tasks show that this family of objectives, particularly the secondary $\mathcal{J}'(\theta)$ and third $\mathcal{J}''(\theta)$ formulations, surpasses the performance of both the GRPO and Supervised Fine-Tuning (SFT) baselines, validating the effectiveness of the proposed theoretical framework.

## 1.1 Literature Review

Reinforcement learning (RL), particularly Deep RL (DRL) and Multi-Agent RL (MARL), has shown significant success in complex sequential decision-making across diverse domains like autonomous navigation (Liu et al., 2025; Bai et al., 2025; Wang et al., 2025), robotic control (Chen & Rojas, 2024; Harish et al., 2024), financial portfolio management (Chen et al., 2023; Hu & Gu, 2024), and game playing (Parisotto et al., 2015; Moniz et al., 2019).

KD is a widely adopted strategy to enhance RL agent efficacy and efficiency, often through teacher-student frameworks. Teacher models can be LLMs guiding Multi-Agent RL (MARL) agents (Liu et al., 2025) or providing rationales (Lee et al., 2024), centralized policies with privileged information (Tseng et al., 2022; Chen & Rojas, 2024), or policies from simpler auxiliary tasks (Harish et al., 2024). The "Actor-Mimic" approach (Parisotto et al., 2015; Moniz et al., 2019) exemplifies multitask learning by training a student to mimic multiple expert teachers, distilling knowledge via policy or feature regression. KD also facilitates managing heterogeneous agents in federated learning (Jiang et al., 2025b), creating distributable policies for multi-robot systems (Wang et al., 2025), and accelerating MARL adaptation (Gao et al., 2021). Distilled knowledge varies from policy outputs (Bai et al., 2025; Jiang et al., 2025a) and features to structural inter-agent relationships (Tseng et al., 2022). KD also helps mitigate catastrophic interference (Zhang et al., 2022) and aids continual learning (Chen & He, 2023).

RL itself is increasingly used to drive the KD process (Xu et al., 2025), for instance, by optimizing teacher strategies (Xu et al., 2025), dynamically weighting multiple teachers based on performance and student-teacher gaps (Yang et al., 2025), adjusting distillation losses (Chen & He, 2023), using LLM-induced rewards (Li et al., 2024a), or guiding automated pruning in model compression (Liu et al., 2023).

Transfer learning is integral to many KD approaches, enhancing generalization. Methods include distilling knowledge from multiple domain-specific teachers into one generalist student (Parisotto et al., 2015; Bai et al., 2025), or concurrently training on main and auxiliary tasks with distillation for behavior transfer (Harish et al., 2024). This is crucial for long-horizon tasks and offline MARL (Tseng et al., 2022). Meta-transfer learning also employs RL to select beneficial source tasks for cross-lingual, low-resource adaptation (He & Fu, 2023). Transfer can be facilitated by privileged teachers distilling to vision-based students using state estimation encoders and weight initialization (Chen & Rojas, 2024). Inverse RL further contributes by estimating reward functions from expert demonstrations to guide policy optimization (Fu et al., 2022).

While the synergy between RL and KD, including MTKD, has shown considerable promise, the specific integration of critic-free RL algorithms, such as GRPO, with guidance from multiple teacher policies remains a relatively underexplored area. The approach that GRPO uses of estimating advantages by comparing rewards of multiple candidate outputs offers unique benefits in terms of stability and efficiency, particularly in complex action spaces like those encountered in LLM and SLM fine-tuning. However, existing literature has not extensively investigated the optimal mechanisms for infusing knowledge from multiple teachers directly into the GRPO learning framework. There is a need for a rigorous theoretical exploration of how to best combine the relative reward optimization of GRPO with the rich, diverse guidance available from an ensemble of teacher policies.

## 2 Preliminaries

Before deriving the GRPO-KD objectives, we establish essential notation and review foundational concepts.

### 2.1 Notation

Within the framework of reinforcement learning, let $s \in \mathcal{S}$ denote a state from the state space $\mathcal{S}$, and $a \in \mathcal{A}$ represent an action from the action space $\mathcal{A}$. We define several distinct policies that govern action selection. The student policy, denoted as $\pi_\theta(a|s)$, is parameterized by a set of parameters $\theta$ and represents the policy being learned or optimized. In contrast, we have a collection of $N$ teacher policies, $\pi_{T_i}(a|s)$ for $i = 1, \ldots, N$, which are assumed to be fixed and provide guidance or expert demonstrations. For regularization purposes, a reference policy $\pi_{ref}(a|s)$ is utilized. The generation of experience data relies on an old policy, $\pi_{old}(a|s)$,

which is typically a previous iteration of the student policy, such as $\pi_{\theta_{old}}$. When sampling from an input state $s$ using this $\pi_{old}(a|s)$, we obtain a group of $K$ outputs $G = \{o_1, \ldots, o_k\}$; for each $k$-th output $o_k$ in this group, a scalar reward $R(o_k)$ is assigned. The group-relative advantage for a specific output $o_k$, denoted $A(s, o_k)$, quantifies the advantage of selecting $o_k$ relative to other outputs within the group $G$ for the given state $s$. The probability ratio for an output $o_k$ is defined as $\rho_k(\theta) = \frac{\pi_\theta(o_k|s)}{\pi_{old}(o_k|s)}$, indicating the relative likelihood of producing output $o_k$ under the current student policy $\pi_\theta$ compared to the policy $\pi_{old}$ used for sampling. Finally, to measure the dissimilarity between the student policy $\pi_\theta$ and the reference policy $\pi_{ref}$, a specific divergence measure, $\mathbb{D}_{KL}[\pi_\theta||\pi_{ref}]$, is employed. It is defined as: $\mathbb{D}_{KL}[\pi_\theta||\pi_{ref}] = E_{s \sim D}\left[\sum_{a \in \mathcal{A}}\left(\frac{\pi_{ref}(a|s)}{\pi_\theta(a|s)} - \log\frac{\pi_{ref}(a|s)}{\pi_\theta(a|s)} - 1\right)\right]$. This definition involves calculating the expectation over states $s$ (drawn from a distribution $D$) of the sum over all actions $a$ of the term $\left(\frac{\pi_{ref}(a|s)}{\pi_\theta(a|s)} - \log\frac{\pi_{ref}(a|s)}{\pi_\theta(a|s)} - 1\right)$. It's worth noting that this functional form, where the per-element divergence is $f(p, q) = \frac{p}{q} - \log\frac{p}{q} - 1$, when summed or integrated over the domain of two distributions $P$ and $Q$, corresponds to the Itakura-Saito (IS) divergence $D_{IS}(P||Q)$. In this context, the expression represents an averaged Itakura-Saito divergence, comparing $\pi_{ref}(\cdot|s)$ to $\pi_\theta(\cdot|s)$ over states $s$ and actions $a$.

## 2.2 Group Relative Policy Optimization (GRPO)

GRPO is an RL algorithm that eliminates the need for an explicit value function. The group-relative advantage for an output $o_k$ from a group $G$ sampled for state $s$ is calculated as:

$$\hat{A}(s, o_k) = \frac{R(o_k) - \mu_R}{\sigma_R + \delta}, \tag{1}$$

where $\mu_R = \frac{1}{K}\sum_{i=1}^{K} R(o_i)$ is the mean reward in the group, $\sigma_R = \sqrt{\frac{1}{K}\sum_{j=1}^{K}(R(o_j) - \mu_R)^2}$ is the standard deviation of rewards, and $\delta$ is a small constant for numerical stability.

The GRPO objective function typically uses a clipped surrogate objective similar to PPO:

$$L_{CL}(\theta) = \hat{\mathbb{E}}_{s, G \sim \pi_{old}}\left[\frac{1}{K}\sum_{k=1}^{K}\min\left(\rho_k(\theta)\hat{A}(s, o_k), \text{clip}(\rho_k(\theta), 1 - \epsilon, 1 + \epsilon)\hat{A}(s, o_k)\right)\right]. \tag{2}$$

GRPO also includes a KL regularization term to constrain policy updates relative to a reference policy $\pi_{ref}$:

$$L_{KL}(\theta) = -\beta\mathbb{E}_{s \sim D}[\mathbb{D}_{KL}(\pi_\theta(\cdot|s)||\pi_{ref}(\cdot|s))]. \tag{3}$$

The full GRPO objective is:

$$\mathcal{J}(\theta) = L_{CL}(\theta) + L_{KL}(\theta). \tag{4}$$

The KL divergence used here is a reverse KL divergence.

## 2.3 Multi-Teacher Knowledge Distillation (MTKD)

KD transfers knowledge from teacher(s) to a student. For multiple teachers $\pi_{T_1}, \ldots, \pi_{T_N}$, a common KD loss is an average of individual KD losses, often using forward KL divergence:

$$L_{KD}(\theta) = \mathbb{E}_{s \sim D'}\left[\sum_{i=1}^{N} w_i(s)\mathbb{D}_{KL}[\pi_{T_i}(\cdot|s)||\pi_\theta(\cdot|s)]\right], \tag{5}$$

where $w_i(s)$ are adaptive, state-dependent weights for each teacher such that $\sum_{i=1}^{N} w_i(s) = 1$. $D'$ is the state distribution for distillation. Forward KL ($D_{KL}(P||Q)$ where $P$ is teacher, $Q$ is student) is "zero-avoiding" or "mean-seeking," encouraging the student to cover all modes of the teacher distribution.

# 3 Theoretical Framework

We now derive the three proposed GRPO-KD objective functions, building upon the GRPO and MTKD frameworks.

## 3.1 Objective 1: Primary GRPO-KD Objective

This method augments the standard GRPO objective with an explicit, separate multi-teacher KD loss term. This approach maintains modularity between relative reward optimization, stability regularization, and teacher mimicry.

The GRPO-KD objective $\mathcal{J}'(\theta)$ objective is constructed by linearly combining three components:

1. The GRPO clipped surrogate objective $L_{CL}(\theta)$ to drive policy improvement based on empirical group-relative rewards.

2. The standard PPO KL regularization term $L_{KL}^{PPO}(\theta) = \mathbb{E}_{s\sim D, a\sim\mathcal{A}}\left[D_{KL}(\pi_\theta||\pi_{ref})\right]$ (Xie et al., 2025) to ensure stability by penalizing deviation from a reference policy $\pi_{ref}$, where the reverse KL is defined as:

$$D_{KL}(\pi_\theta||\pi_{ref}) = \sum_{s\sim D}\sum_{a\sim\mathcal{A}} \pi_\theta(a|s) \log\frac{\pi_\theta(a|s)}{\pi_{ref}(a|s)}. \tag{6}$$

3. A multi-teacher KD loss term, $L_{KD}(\theta)$, using forward Hinton's KL divergence (Hinton et al., 2015) to encourage the student $\pi_\theta$ to capture the breadth of behaviors from the ensemble of teacher policies $\pi_{T_i}$. The Hinton KL divergence is defined as:

$$D_H(\pi_{T_i}||\pi_\theta) = \tau^2 \sum_{a\in\mathcal{A}} \zeta(z_{T_i},\tau)_a \log\left(\frac{\zeta(z_{T_i},\tau)_a}{\zeta(z_\theta,\tau)_a}\right), \tag{7}$$

where $\tau$ is the temperature, and $\zeta(z,\tau)_a$ is the temperature-scaled softmax function for a specific action $a$. The softmax function is defined as:

$$\zeta(z,\tau)_a = \frac{e^{z_a/\tau}}{\sum_{j\in\mathcal{A}} e^{z_a/\tau}}, \tag{8}$$

such that $z_a$ is the logit for action $a$, $z_{T_i}$ and $z_\theta$ are the logit vectors produced by the teacher and student models, respectively, for a given state $s$.

The combined objective function is:

$$\begin{aligned}
\mathcal{J}'(\theta) =& \hat{\mathbb{E}}_{s,G\sim\pi_{old}}\left[\frac{1}{K}\sum_{k=1}^{K}\min\left(\rho_k(\theta)\hat{A}(s,o_k), \text{clip}(\rho_k(\theta), 1-\epsilon, 1+\epsilon)\hat{A}(s,o_k)\right)\right] \\
& - \beta\mathbb{E}_{s\sim D}\left[D_{KL}(\pi_\theta(\cdot|s)||\pi_{ref}(\cdot|s))\right] + \lambda\mathbb{E}_{s\sim D'}\left[\sum_{i=1}^{N} w_i(s)D_H(\pi_{T_i}(\cdot|s)||\pi_\theta(\cdot|s))\right]
\end{aligned} \tag{9}$$

which combines three key terms: the first, $L_{CL}(\theta)$, represents the primary policy optimization; the second term incorporates PPO stability regularization, controlled by the hyperparameter $\beta$; and the third term introduces a MTKD loss, where the student policy $\pi_\theta$ is encouraged to match an adaptively weighted ($w_i(s)$) average action probability distribution of the teacher policies $\pi_{T_i}$, with the hyperparameter $\lambda$ governing the strength of this distillation and the state distribution $D'$ for the KD term being tailorable for effective distillation.

### 3.2 Objective 2: Modified KL Regularization GRPO-KD

This approach integrates teacher guidance more directly by modifying the original KL regularization term of GRPO. Instead of penalizing deviation from $\pi_{ref}$ with the IS divergence $\mathbb{D}_{KL}[\pi_\theta||\pi_T]$, the term penalizes deviation from a weighted average of the teacher policies with the Hinton KL divergence $D_H(\pi_\theta||\pi_T)$.

The core idea is to replace the reference policy $\pi_{ref}$ in the KL regularization term with a target derived from the teacher ensemble $\pi_{T_i}$. This uses a forward Hinton KL divergence $(D_H(\pi_\theta||\pi_T))$.

The KL regularization term becomes:

$$L_H(\theta) = -\beta' \mathbb{E}_{s \sim D} \left[ \sum_{i=1}^{N} w_i(s) D_H(\pi_\theta(\cdot|s)||\pi_{T_i}(\cdot|s)) \right] \tag{10}$$

We do note that this formulation suggests $D_H[\pi_\theta(\cdot|s)||\pi_{T_i}(\cdot|s)]$ for each teacher, and then these are weighted. This implies the student is regularized towards each teacher individually, with the overall strength modulated by $w_i(s)$ and $\beta'$.

The full objective $L''(\theta)$ is then:

$$\begin{aligned}
\mathcal{J}''(\theta) =& \hat{\mathbb{E}}_{s, G \sim \pi_{old}} \left[ \frac{1}{K} \sum_{k=1}^{K} \min \left( \rho_k(\theta) \hat{A}(s, o_k), \mathrm{clip}(\rho_k(\theta), 1 - \epsilon, 1 + \epsilon) \hat{A}(s, o_k) \right) \right] \\
& - \beta' \mathbb{E}_{s \sim D} \left[ \sum_{i=1}^{N} w_i(s) D_H[\pi_\theta(\cdot|s)||\pi_{T_i}(\cdot|s)] \right],
\end{aligned} \tag{11}$$

where $\beta'$ is a hyperparameter controlling the strength of both stability regularization and distillation. The weights $w_i(s)$ determine the relative influence of each teacher policy in the regularization term.

This formulation attempts a tighter coupling of distillation with the stabilization mechanism. The forward Hinton KL divergence encourages the student policy $\pi_\theta$ to be "mode-seeking," focusing on regions of high probability under the teacher policies. However, this conflates the roles of stability (preventing large updates from $\pi_{old}$) and teacher guidance, which can make tuning $\beta'$ and designing $w_i(s)$ more challenging.

### 3.3 Objective 3: Teacher-Augmented Advantage GRPO-KD

This advanced strategy aims for a deeper integration of teacher knowledge by directly modifying the advantage calculation of GRPO. It introduces a "Teacher Agreement Score" (TA) that augments the group-relative advantage.

The $TA(o_k, s)$ quantifies the consensus among the teacher ensemble for a output $o_k$ of the student in state $s$. It is defined as the natural logarithm of the weighted average probability assigned to $o_k$ by the teachers:

$$TA(o_k, s) = \log \left( \sum_{i=1}^{N} w_i(s) \pi_{T_i}(o_k|s) \right), \tag{12}$$

where $\pi_{T_i}(o_k|s)$ is the probability of output $o_k$ (which could be a sequence of actions $(a_1, \ldots, a_L)$, so $\pi_{T_i}(o_k|s) = \prod_{l=1}^{L} \pi_{T_i}(a_l|s, a_1, \ldots, a_{l-1})$) according to teacher $i$, and $w_i(s)$ are the state-dependent adaptive weights for each teacher, $\sum w_i(s) = 1$. A higher $TA(o_k, s)$ indicates stronger teacher agreement for output $o_k$.

The standard group-relative advantage $A(s, o_k)$ is augmented by the $TA(o_k, s)$ to form the normalized Teacher-Augmented Advantage Function for stability:

$$\hat{A}'(s, o_k) = \hat{A}(s, o_k) + \gamma \frac{TA(o_k, s) - \mu_{\mathrm{TA}}}{\sigma_{\mathrm{TA}} + \delta}, \tag{13}$$

where $\gamma$ is a hyperparameter controlling the strength of the normalized teacher agreement bonus. This directly incorporates teacher preferences into the advantage signal that is being optimized.

| Model | GSM8K | MathInstruct |
|---|---|---|
| Flan-T5-base | 4.02% | 38.55% |
| Flan-T5-small | 2.20% | 10.85% |
| Flan-T5-small + SFT | 1.67% | 22.16% |
| Flan-T5-small + GRPO | 1.97% | 1.26% |
| Flan-T5-small + $\mathcal{J}'$ | 2.46% | 24.09% |
| Flan-T5-small + $\mathcal{J}''$ | **2.58%** | 24.11% |
| Flan-T5-small + $\mathcal{J}'''$ | 2.46% | **26.22%** |

Table 1: Comparison between the proposed GRPO-KD objectives and the base methods

The $\mathcal{J}'''(\theta)$ objective uses this new $\hat{A}'(s, o_k)$ in its surrogate objective component, while retaining the structure of the primary $\mathcal{J}'$ (i.e., separate GRPO KL regularization and an explicit KD loss term):

$$
\mathcal{J}'''(\theta) = \hat{\mathbb{E}}_{s,G \sim \pi_{old}} \left[ \frac{1}{K} \sum_{k=1}^{K} \min \left( \rho_k(\theta) \hat{A}'(s, o_k), \mathrm{clip}(\rho_k(\theta), 1 - \epsilon, 1 + \epsilon) \hat{A}'(s, o_k) \right) \right]
$$
$$
- \beta \mathbb{E}_{s \sim D} \left[ D_{KL}(\pi_\theta(\cdot|s) || \pi_{ref}(\cdot|s)) \right] + \lambda \mathbb{E}_{s \sim D'} \left[ \sum_{i=1}^{N} w_i(s) D_H [\pi_{T_i}(\cdot|s) || \pi_\theta(\cdot|s)] \right].
$$
(14)

This formulation represents the most intrinsic infusion of teacher guidance. By modifying the advantage estimate $\hat{A}$, the preferences of the teacher ensemble directly shape the reward landscape perceived by the RL agent. The $TA$ score acts as a teacher-defined pseudo-reward or preference signal. This allows the primary learning signal of the agent (the advantage) to be directly influenced by teacher consensus, potentially leading to more aligned and efficient learning, especially when the extrinsic reward $R(o_k)$ is sparse or noisy. The other terms (KL regularization against $\pi_{ref}$ and the explicit KD loss) provide additional stability and direct policy mimicry, respectively.

## 4 Experimental Results

The primary results of our investigation are presented in Table 1. This table details the performance of our proposed GRPO-KD objective functions ($\mathcal{J}'$, $\mathcal{J}''$, and $\mathcal{J}'''$) against several baseline methods. Due to resource constraints, the Flan-T5 family of models was chosen to test out the GRPO-KD framework. The experiments were conducted using the SLM Flan-T5-small as the student model, fine-tuned on the GSM8K (Cobbe et al., 2021) and MathInstruct (Yue et al., 2023) reasoning benchmarks. The GSM8K dataset is used for the CoT reasoning task, while the MathInstruct dataset is used for the multiple-choice question answering task. The performance of the larger Flan-T5-base model is included as a reference for the capabilities of a teacher model. The code used for the experimentation can be found on GitHub [1].

### 4.1 Performance on Reasoning Tasks

Our analysis of these results yields three key observations.

First, conventional fine-tuning methods demonstrate significant limitations for these complex reasoning tasks. On the MathInstruct dataset, the standard GRPO algorithm, lacking teacher guidance, exhibited a severe performance degradation, with accuracy dropping to 1.26% from the base model's 10.85%. This underscores the inherent instability of applying critic-free RL to complex, high-dimensional policy spaces without a robust guiding signal. While SFT provided a substantial improvement to 22.16% on MathInstruct, its efficacy did not generalize to the GSM8K task, where it resulted in a performance decrease to 1.67%. This highlights the inadequacy of standard fine-tuning approaches for robustly enhancing reasoning capabilities.

---

[1]GitHub experimentaion code link: `https://github.com/[REDACTED_FOR_ANONYMITY]`

Second, our proposed GRPO-KD framework consistently and significantly outperforms all baseline methods. As evidenced in Table 1, all three GRPO-KD formulations successfully overcome the instability of the standard GRPO algorithm. On GSM8K, where baselines failed to improve performance, our methods increased accuracy to 2.46% ($\mathcal{J}'$, $\mathcal{J}'''$ J) and 2.58% ($\mathcal{J}''$), respectively. The improvements were even more pronounced on MathInstruct. All GRPO-KD objectives surpassed the strong SFT baseline, with $\mathcal{J}'$ achieving 24.09%, $\mathcal{J}''$ achieving 24.11%, and $\mathcal{J}'''$ achieving 26.22%. This confirms that integrating multi-teacher knowledge distillation provides the necessary guidance and regularization to make GRPO an effective optimization strategy for SLMs.

Third, the method of integrating teacher knowledge influences final performance, with deeper integration proving highly effective. While all proposed objectives were successful, we observe that the optimal formulation varied by task. On GSM8K, the $\mathcal{J}''$ objective, which replaces the KL regularization term with a teacher-derived target, performed best. On the MathInstruct benchmark, the $\mathcal{J}'''$ objective yielded the highest performance. This formulation represents the deepest level of integration by directly augmenting the advantage signal with a Teacher Agreement Score. Its superior performance suggests that directly embedding teacher consensus into the core learning signal of the RL agent is a particularly potent strategy for complex reasoning tasks where the reward landscape is sparse or difficult to navigate.

In summary, the experimental results robustly validate the theoretical framework proposed in this paper. The consistent and significant performance gains demonstrate that augmenting a critic-free RL algorithm like GRPO with multi-teacher knowledge distillation is an effective and stable approach for fine-tuning SLMs on challenging reasoning benchmarks.

## 4.2   Training Dynamics

To further understand the performance differences reported in Table 1, we examined the training dynamics of the models. The log-scaled power law decay functions fitted to the loss curves for the GSM8K and MathInstruct datasets are plotted in Figure 2 and Figure 3, respectively. These plots provide a clear visual representation of the learning stability and convergence behavior of the standard GRPO baseline versus our proposed GRPO-KD objectives.

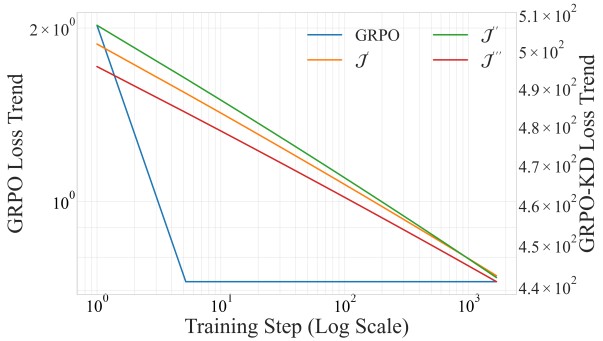
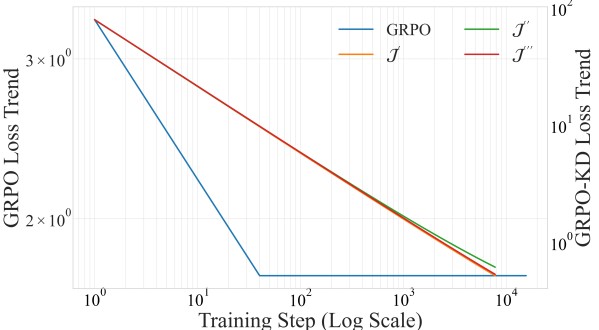

Figure 2: Loss convergence for GRPO and GRPO-KD on GSM8K.

Figure 3: Loss convergence for GRPO and GRPO-KD on MathInstruct.

A key observation from both figures is the deceptive convergence of the standard GRPO baseline. In both training runs, the loss for the GRPO model (blue line) drops precipitously within the initial training steps, quickly reaching a floor and flatlining for the remainder of the process. This rapid "convergence" to a minimal loss value, when juxtaposed with the poor final accuracies in Table 1, indicates that the policy is not engaging in meaningful learning. Instead, it is likely collapsing into a simple, sub-optimal policy that satisfies the immediate objective but fails to generalize or solve the underlying reasoning task. This behavior is a classic symptom of instability in reinforcement learning, where the agent finds a trivial solution that yields a low loss but poor performance.

In stark contrast, the loss curves for all GRPO-KD formulations ($\mathcal{J}'$, $\mathcal{J}''$, and $\mathcal{J}'''$) exhibit the characteristics of a stable and productive learning process. Their loss values start higher than the GRPO baseline's initial drop but show a consistent, steady descent across thousands of training steps. This gradual and sustained reduction in loss, which correlates with their superior performance in Table 1, demonstrates that the student model is continuously refining its policy and genuinely learning the complexities of the reasoning tasks. The knowledge distillation component acts as a crucial regularizer, guiding the learning process and preventing the policy collapse observed in the unguided GRPO baseline.

In conclusion, the visual evidence presented in the loss plots strongly corroborates our primary findings. The unstable training dynamics of standard GRPO explain its failure to produce effective policies for complex reasoning. The stable, gradual convergence of the GRPO-KD models confirms that the integration of multi-teacher knowledge distillation is essential not only for achieving high performance but also for ensuring a robust and reliable training process.

## 5 Conclusion

In this paper, we addressed the challenge of effectively fine-tuning language models for complex reasoning tasks using critic-free reinforcement learning. While algorithms like Group Relative Policy Optimization (GRPO) offer benefits in efficiency and stability by forgoing a value function, their direct application can lead to policy collapse and sub-optimal performance especially on SLMs. To overcome this, we introduced a novel theoretical framework that synergizes GRPO with Multi-Teacher Knowledge Distillation (MTKD).

Our primary contribution is the formalization of a new family of objective functions, termed GRPO-KD, which integrate guidance from an ensemble of expert teacher policies directly into the learning process. We proposed three distinct formulations, each offering a progressively deeper level of integration: an additive distillation loss ($\mathcal{J}'$), a teacher-guided KL regularization term ($\mathcal{J}''$), and a novel Teacher-Augmented Advantage function ($\mathcal{J}'''$).

Our experimental evaluation on the GSM8K and MathInstruct benchmarks provided compelling evidence for the efficacy of our approach. The results demonstrated that all three GRPO-KD objectives substantially outperform baseline methods, including standard GRPO and Supervised Fine-Tuning. Furthermore, analysis of the training dynamics revealed that our framework resolves the inherent instability of standard GRPO, transforming its rapid but deceptive convergence into a stable and meaningful learning process. The superior performance of the $\mathcal{J}'''$ objective on the MathInstruct task suggests that directly embedding teacher consensus into the advantage signal of the agent is a particularly potent strategy.

This work validates that the fusion of critic-free RL and knowledge distillation is a powerful paradigm for developing capable, small-scale language models. Looking forward, this research opens several promising avenues for future inquiry. A comprehensive empirical evaluation across a wider array of tasks and model architectures is a key next step. Future work should also focus on developing more sophisticated dynamic weighting mechanisms ($w_i(s)$) for the teacher policies, potentially based on measures of teacher competence or student uncertainty. Finally, exploring the theoretical convergence properties of the proposed objectives and extending the GRPO-KD framework to other domains, such as multi-agent systems and other generative tasks, represent valuable directions for further investigation.

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
