# OpenReview forum: "Multi-Teacher Knowledge Distillation Augmented Group Relative Policy Optimization"
_TMLR — Rejected by TMLR_

### Review · Reviewer_FHuB · 2025-07-23

**Summary Of Contributions:**

The paper proposes to compare different losses in critic-free RL, using GRPO, for knowledge distillation of small language models. More specifically, it compares three different variations of the GRPO objective for two reasonning tasks, using Flan-T5. The proposal looks to overperforms classical supervised finetuning, but the contribution lacks providing sufficient clues of performance.

**Additional Comments:**

Minor comments :
- Fig1 is not discussed. It arrives too early to be fully understandable, and should be placed after the notation part.
- Authors should better explain their KL term detailed in the end of 2.1, as an unbiased estimator proposed in (Schulman, 2020)  (rather than talking about IS divergence without not enough justification)
- the evaluation metric is not specified
- the reward is not specified

**Audience:**

No

**Audience Explanation:**

Findings can be potentially interesting but the paper is not sufficiently detailled to be useful in the current form.

**Broader Impact Concerns:**

.

**Claims And Evidence:**

No

**Claims Explanation:**

- The proposals of the paper are not sufficiently justified and detailled. Simply giving three different losses without any justification is not enough to me. The section is called "theoretical framework", but it does not include any theoretical discussion about the choices made in the different variants. Intuitive arguments are also not sufficient. And rather straightforward. The first proposed objective simply adds a regularization term (MTKD) that aims at steering the model towards actions of the teacher(s), that is very similar to approaches such as the one in [1] for instance, which is far from novel. The second objective replaces the classical regularization term in GRPO by a reversed MTKD, that aims at preventing leaving actions from the teacher(s). The third one is similar to the first one, while augmenting the normalized advantage with an aggregated likelihood of selected actions from the teachers' perspective. At least, I would suggest to analyze what means this additional term and in what sense it differs from the MTKD term (as it is similar to a way to estimate KL). Writting is not satisfactory in general. Authors should unify the three objectives in a single formula. And discuss more thoroughly the different components (reverse vs forward, etc) that constitute their contribution. The contribution is not fully clear. Related Work is also not enough structured and does not sufficiently position the work w.r.t. to the litterature.

- The paper claims coping with multi-teacher distillation but only experiments with a single teacher.

- Experiments are not sufficiently detailled to be reproducible and to fully validate findings. For instance, nothing is said about the hyper-parameters used in the approaches. I suspect that the beta parameter has a very strong impact on the GRPO baseline. Its impact should be studied. Also the temperature parameter used in the hinton KL divergence is not discussed.

- The authors claim inspect the use of critic-free RL for distillation. But do not compare with classical approaches using critic networks. They claim that GRPO induces more stability and efficiency. But this is not fully clear to me as GRPO requires many rollouts to estimate advantage. The paper should include a PPO baseline.

- Also a baseline that uses more classical distillation approaches would have been useful. For instance an approach that imitate logits from the LLM at each step of the generation (which looks very close from the third proposed objective without normalization and reward).

- The way adavantage is estimated is not fully specified. Authors say that o_k can be a sequence of actions. Is o_k always the full generated text from the prompt (that stands for s in that setting) in the experiments ? If yes, I suspect a very high variance for the IS ratio \rho(o_k). If not, how authors estimate advantage for their third objective where we get per-step rewards. Do they use the process described in section "Process Supervision RL with GRPO" from the original GRPO paper (Shao et al., 2024) ? For other objectives the signal is the same for any step, right ?

- Authors experiment on hard tasks on which models obtain rather low scores. It would be useful to experiment on easier tasks, as comparing for instance 4% accuracy with 6% accuracy may be not really significant. Statistical significance of results should also be given.


[1] Vilouras, Konstantinos, et al. "Group distributionally robust knowledge distillation." International Workshop on Machine Learning in Medical Imaging. Cham: Springer Nature Switzerland, 2023.

**Requested Changes:**

- Provide more justifications of the archiectural choices
- Provide more baselines, ablations and experimental details

---

### Review · Reviewer_66oB · 2025-07-30

**Summary Of Contributions:**

This paper addresses the important problem of fine-tuning Small Language Models (SLMs) combining RL with multi-teacher distillation to stabilize training. To this end, they modify the KL regularization in RL, and also investigate a Teacher Agreement Score to be included in the advantage. They implement their ideas on Flan-T5-small, and show some slight improvements on GSM8k and MathInstruct.

**Audience:**

No

**Audience Explanation:**

In the absolute, the topic of combining distillation and RL for finetuning of SLMs is a super promising research direction that would interest many researchers in the TLMR audience. Unfortunately, the paper in its current form does not provide significant insights, and requires a major revision before re-evaluation. As stated above, the submission suffers from a disconnect between its central claims and its evaluation, clear experimental flaws undermining the validity of its conclusions.

**Broader Impact Concerns:**

No broad impact concern.

**Claims And Evidence:**

No

**Claims Explanation:**

The claims are usually not convincing and not supported by evidence.

* Multi-teacher approach though a single-teacher experiment. The most significant flaw is the contradiction between the paper's title and central theme ("Multi-Teacher Distillation") and the experiment, which appears to use only a single teacher model (Flan-T5-base). Action: Conduct proper multi-teacher experiments.

* Clarity on the paper through thorough ablation. The paper presents three competing objective functions without a clear organization and recommendation. They are actually all different variants with some components added or removed. Action: Therefore, to clarify the overall message of the paper, I would recommend present one single objective, and in the experimental section investigate which part is actually useful with proper ablation.

* Empirical significance: baseline. The experimental setup is weak, making the results difficult to trust or reproduce. Notably, the standard GRPO baseline implementation seems non-functional (its loss collapses instantaneously and it reduces GSM8k performances). Moreover, the paper lacks a proper distillation baseline without any RL. Minor: The code could be made anonymous and provided.

* Empirical significance: overall performance. The results do not demonstrate that the proposed method is practically useful. Indeed, the absolute performance on GSM8K (2.58% at best) is extremely low, even for the teacher (4.02%). Action: To demonstrate practical utility, take a larger teacher and potentially a better/more recent student. If compute is a constraint, consider using LORA or simpler dataset.

* Empirical significance: statistical analysis. Results are reported as single-run values without error bars or standard deviations. This makes it impossible to assess whether the small reported gains are statistically significant or random noise, as the difference between the best and worst strategies are +0.12. Action: Re-run all experiments over multiple random seeds (e.g., 3-5) and report both mean and standard deviation for all results in Table 1.

## More Minors

* Consider simplifying the terminology. For instance, rather than explaning in details "Hinton's KL divergence" with new notations, this can be summarized by saying: "KD with temperature" and later detailed in Appendix if you think it's necessary.

* Please add a dedicated ablation study section to detail and investigate the impact of key hyperparameters.

* Unclear Figures 2 and 3. The composite loss plots are hard to understand (except to see GRPO loss collapse!). It would be more informative to plot the individual training curves (reward, KL divergence, distillation loss) separately.

* Potential error in Equations 9 and 14. There appears to be a sign error in Equations 9 and 14, where the distillation loss term is added to an objective being maximized, while such loss should be subtracted.

* "the main contribution of this paper is the formalization of a theoretical basis extending GRPO with multi-teacher KD... We believe this is the first attempt at merging these two concepts together." Reward-guided distillation is an active area of research, see for example this recent "Advantage-guided distillation for prefer-ence alignment in small language models" (ICLR 2025). More generally, a better related work would be required for pubication.

**Requested Changes:**

See the "action" items in the answers above.

---

### Review · Reviewer_r94C · 2025-08-27

**Summary Of Contributions:**

This paper addresses the challenge of fine-tuning Small Language Models (SLMs) on complex reasoning tasks using critic-free reinforcement learning. The authors identify that Group Relative Policy Optimization (GRPO), while efficient, suffers from training instability and poor performance when applied directly to SLMs without expert guidance. To solve this, they introduce GRPO-KD, a novel framework that synergizes GRPO with Multi-Teacher Knowledge Distillation (MTKD). The core contribution is the formalization of a family of three objective functions (J', J'', and J'''), each representing a progressively deeper integration of teacher guidance into the GRPO learning process. These methods range from adding an explicit distillation loss term to directly modifying the advantage calculation with a novel "Teacher Agreement Score." Through experiments on the GSM8K and MathInstruct reasoning benchmarks, the authors demonstrate that all three GRPO-KD formulations not only stabilize the training process but also significantly outperform standard GRPO and Supervised Fine-Tuning (SFT) baselines.

**Audience:**

Yes

**Audience Explanation:**

GRPO is a prominent learning algorithm for reasoning models and this paper tries to solve the well-known instability problem.

**Broader Impact Concerns:**

None.

**Claims And Evidence:**

No

**Claims Explanation:**

The central claims are that (1) standard GRPO is unstable for fine-tuning SLMs on complex tasks, and (2) augmenting it with multi-teacher knowledge distillation (GRPO-KD) resolves this instability and significantly improves performance.  The catastrophic failure of the standard GRPO baseline on the MathInstruct dataset, where accuracy drops to 1.26% from a 10.85% base (Table 1), provides powerful and convincing evidence for the initial claim of instability. Table 1 clearly shows that all three proposed GRPO-KD objective functions substantially outperform both the unstable GRPO baseline and the stronger Supervised Fine-Tuning (SFT) baseline across both GSM8K and MathInstruct datasets. This directly supports the primary claim of the paper's effectiveness.

While the results themselves are convincing, the evidence is not fully clear or accurate because the experimental methodology is missing crucial details. This prevents the results from being fully verifiable or reproducible. Specifically:

- The paper never specifies the identity, number, or configuration of the teacher models used in the "multi-teacher" ensemble. This is a critical omission, as the quality and diversity of teachers are fundamental to the success of the method.
- The "adaptive, state-dependent weights" wi(s) are a core component of all three proposed objective functions, yet the paper provides no information on how these weights are calculated or updated.
- Key hyperparameters that control the balance between RL optimization and knowledge distillation (λ, β', and γ) are not reported.

**Requested Changes:**

Based on my review, I recommend the following changes. The first set of changes is critical for ensuring the work is reproducible and the claims are fully verifiable. The second set would significantly strengthen the paper's contribution and impact.

**Critical For Acceptance (for Reproducibility and Clarity)**

1.  The paper's reproducibility is severely hampered by the lack of detail about the teacher models. Please add a dedicated section or paragraph in the experimental setup that clearly states:
    *   The exact models used as teachers (e.g., Flan-T5-base, Llama-2-7B, etc.).
    *   The number of teachers in the ensemble (`N`).
    *   The rationale for choosing this specific set of teachers. Was diversity a factor? The current mention of Flan-T5-base as a "reference for the capabilities of a teacher model" is ambiguous and insufficient.

2.  The adaptive weights `wi(s)` are a core component of all three proposed objective functions (Equations 9, 11, 12, 14), but their calculation is never explained. This is a fundamental omission. Please explicitly define how `wi(s)` is determined. For example:
    *   Are the weights uniform (e.g., `1/N`)?
    *   Are they learned parameters? If so, how are they trained?
    *   Are they calculated based on a heuristic, such as teacher confidence or student uncertainty?

3.  The crucial hyperparameters that control the influence of the teacher guidance: `λ` in `J'`, `β'` in `J''`, and `γ` in `J'''`, are not mentioned in the experimental results. Please report the specific values used to obtain the results in Table 1. This is essential for anyone trying to reproduce or build upon this work.

**Good-To-Haves**

1.   The paper emphasizes the "multi-teacher" aspect of the framework. To empirically validate the necessity of this component, please consider adding a baseline that uses only a single teacher (e.g., GRPO + single-teacher KD). This would demonstrate whether the performance gains are attributable to KD in general or specifically to the multi-teacher ensemble.

2.  The paper motivates GRPO as a "critic-free" method but shows it to be highly unstable on its own. Including a widely-used critic-based algorithm like Proximal Policy Optimization (PPO) as a baseline would provide valuable context. It would help answer: Is the observed instability a general problem for RL on these tasks, or is it specific to the critic-free approach? How does a standard PPO+KD setup compare?

3.  To further enhance the paper's contribution, an analysis of the model's sensitivity to the key hyperparameters (`λ`, `β'`, `γ`) would be very valuable. This would provide readers with a deeper understanding of the method's robustness and offer guidance for applying it to new tasks. This would be particularly insightful for `J''`, where the authors themselves note the potential difficulty in tuning the conflated `β'` parameter.

---

### Decision · Action_Editor_7CoT · 2025-10-08

**Recommendation:** Reject

**Audience:**

No

**Audience Explanation:**

The subject of the paper is itself a good match for TMLR, but seeing the discrepancy between the claims and the actual results, I don't think it would be of a great interest to the community.

**Claims And Evidence:**

No

**Claims Explanation:**

The paper studies a combination of RL and policy distillation from a teacher model. They introduce for this a new algorithm, based on GRPO, that modifies the KL regularization term in the algorithm to introduce a distillation loss. They also explain how to use their algorithm with multiple teachers thanks to a teacher agreement score. Then, their method is evaluated with open models on standard reasoning benchmarks.

There are several issues with the paper that were pointed out by the reviewers. The authors have not answered to any concerns from the reviewers, so I will not go into details, but will summarize the main issues:

- Multiple teachers / single teacher : one of the main claim of the paper is to allow distillation from multiple teachers at the same time, however, this setup is actually never tested empirically in this work, and the experiments are all done with one teacher only. This makes the framing of the paper deceptive, as the multiple teacher setup is a key argument from the work. This was pointed out by all the reviewers.

- Empirical significance: the results presented in the paper do not seem to be significant enough to support the claim of the authors. Notably, it was pointed out by several reviewers that the chosen benchmark (GSM8k) is not suited fro the models: most of the experiment presented in the paper perform quite poorly on the task, and what we are comparing is most likely noise rather than actual improvement. It was also pointed out that the baseline seemed flawed, as, for example, the only-GRPO baseline did not improve the scores on GSM8k.

Overall, the paper, while studying a clearly important problem for LM post-training, and offering lcear novelty, does not match its claims with proper results. The setup it claims to study is only suggested theoretically, but not tested, and the chosen tasks and baseline do not allow for a proper evaluation of the methods.

Since the authors have not answered to the reviews, and the reviewers unanimously recommend rejection, I recommend rejecting the paper.